# Awake Craniotomy for Gliomas in the Non-Dominant Right Hemisphere: A Comprehensive Review

**DOI:** 10.3390/cancers16061161

**Published:** 2024-03-15

**Authors:** Dilshod Muhammadvalievich Mamadaliev, Ryuta Saito, Kazuya Motomura, Fumiharu Ohka, Gianluca Scalia, Giuseppe Emmanuele Umana, Alfredo Conti, Bipin Chaurasia

**Affiliations:** 1Department of Neurosurgery, Nagoya University Hospital, Nagoya 466-8550, Japan; dr.mdm@mail.ru (D.M.M.); ryuta.saito@nngoyansurg.com (R.S.); k.motomura@nsurg.com (K.M.); f.ohka@nsurg.com (F.O.); 2Department of Skull Base Neurosurgery, Republican Specialized Scientific-Practical Medical Center of Neurosurgery of Uzbekistan, Tashkent 100140, Uzbekistan; 3Neurosurgery Unit, Department of Head and Neck Surgery, ARNAS Garibaldi, 95123 Catania, Italy; gianluca.scalia@outlook.it; 4Department of Neurosurgery, Trauma and Gamma Knife Center, Cannizzaro Hospital, 95126 Catania, Italy; umana.nch@gmail.com; 5IRCCS Istituto delle Scienze Neurologiche di Bologna, via Altura 3, 40139 Bologna, Italy; 6Dipartimento di Scienze Biomediche e Neuromotorie (DIBINEM), Alma Mater Studiorum Università di Bologna, 40123 Bologna, Italy; 7Department of Neurosurgery, Neurosurgery Clinic, Birgunj 44300, Nepal; trozexa@gmail.com

**Keywords:** awake craniotomy, intraoperative monitoring, low-grade glioma, theory of mind, medial ventral premotor cortex, prefrontal cortex

## Abstract

**Simple Summary:**

This article aims to comprehensively review the current literature on the benefits of awake craniotomy in gliomas of the non-dominant right hemisphere. A systematic review was conducted using the PubMed and ScienceDirect databases. The literature search identified 74 sources, including original articles, books, monographs, and review articles showing that awake surgery for non-dominant-hemisphere gliomas improves patients’ outcomes.

**Abstract:**

Awake surgery has become a standard practice for managing diffuse low-grade gliomas (LGGs), particularly in eloquent brain areas, and is established as a gold standard technique for left-dominant-hemisphere tumors. However, the intraoperative monitoring of functions in the right non-dominant hemisphere (RndH) is often neglected, highlighting the need for a better understanding of neurocognitive testing for complex functions in the right hemisphere. This article aims to comprehensively review the current literature on the benefits of awake craniotomy in gliomas of the non-dominant right hemisphere. A systematic review was conducted using the PubMed and ScienceDirect databases with keywords such as “right hemisphere”, “awake surgery”, “direct electrical brain stimulation and mapping”, and “glioma”. The search focused on anatomical and surgical aspects, including indications, tools, and techniques of awake surgery in right cerebral hemisphere gliomas. The literature search identified 74 sources, including original articles, books, monographs, and review articles. Two papers reported large series of language assessment cases in 246 patients undergoing awake surgery with detailed neurological semiology and mapping techniques, while the remaining studies were predominantly neuroradiological and neuroimaging in nature. Awake craniotomy for non-dominant-hemisphere gliomas is an essential tool. The term “non-dominant” should be revised, as this hemisphere contributes significantly to essential cognitive functions in the human brain.

## 1. Introduction

Awake surgery has become a common strategy in the contemporary management of cerebral gliomas in eloquent brain areas in many neurosurgical institutions [1,2]. It serves as an armamentarium for achieving the basic aims of neurosurgical oncology, namely, maximizing the extent of tumor resection (EOR) while preserving neurological functions [3]. An increased EOR is associated with improvements in the overall survival rate for patients with both low- and high-grade gliomas [4,5,6,7,8,9,10,11,12,13,14]. Initially, awake procedures were predominantly employed in surgeries for the left cerebral hemisphere, considered an eloquent hemisphere, while the right hemisphere was often neglected due to its perceived lesser importance in terms of functionality. However, the notion of the right hemisphere having only a “minor function” is now being challenged, as subtle postoperative neuropsychological complications in cognitive and behavioral functions have been reported after right hemisphere surgery [1,2]. Key cognitive functions such as visuospatial and social cognitions are determined by the non-dominant right hemisphere. Visuospatial cognition involves spatial awareness, perception, and the representation of space, while social cognition encompasses functions like empathy, theory of mind (TOM), nonverbal language, facial expression recognition, and emotional prosody. The dogmatic concept of hemispheric dominance localization and the fixed understanding of the right hemisphere as non-dominant have led many neurosurgeons to underestimate its functional significance. Cases of right-hemispheric lesions are often operated upon under general anesthesia, based on the perceived low risk of permanent neurological sequelae, with a focus on monitoring motor functions. Despite the importance of right hemisphere functions, there is a paucity of literature on the use of cortical and subcortical mapping on the right hemisphere compared to linguistic mapping in the left hemisphere [15,16,17,18,19,20,21,22,23,24,25,26,27,28,29,30,31,32,33]. This lack of interest may stem not only from the underestimation of the right hemisphere’s contribution to cognitive function but also from the intricacies of functional anatomy and the challenges involved in using traditional bedside tasks in awake-surgical situations. In low-grade gliomas (LGGs), where neuroplasticity is apparent before surgery and may persist during and after surgery, relying solely on anatomical criteria for language functions may be unreliable [34,35,36,37,38,39]. A clear understanding of the functions and symptomatology of the right cerebral hemisphere, along with an analysis of the applicability of intraoperative testing methods, will enhance neurosurgeons’ utilization of awake surgery in right-hemispheric lesions. Thus, we undertook a comprehensive review of the current literature on the use of awake surgery in right-hemispheric lesions.

## 2. Materials and Methods

The objective of our clinical review was to enhance understanding and elucidate the methods employed for the intraoperative evaluation of right-hemispheric functions by comprehensively reviewing the most reliable intraoperative function-targeted tests.

Literature Search: This review utilized the PubMed and ScienceDirect databases for data retrieval. The search strategy involved the keywords “right hemisphere”, “awake surgery”, “direct electrical brain stimulation and mapping”, and “glioma”. The literature search was conducted from November to December 2023. The initial screening involved eliminating duplicate papers, followed by filtering based on titles, abstracts, and full-text assessment to ensure the inclusion of relevant studies. Particular attention was given to papers focusing on anatomical and surgical aspects, evaluating the nuances of indications, tools, and techniques specific to awake surgery in gliomas located in the right cerebral hemisphere. All selected papers were required to be written in English.

Inclusion criteria: papers meeting the following criteria were included in our review:Relevance to awake surgery in gliomas of the right cerebral hemisphere.Presentation of anatomical and surgical considerations.Discussion of indications for awake surgery in the right hemisphere.Exploration of tools and techniques used during intraoperative assessment.

Exclusion criteria: papers were excluded if they did not meet the inclusion criteria or if they were not written in English.

Data extraction: data extraction was systematically carried out, focusing on key aspects such as:Anatomical considerations specific to the right hemisphere.Surgical techniques employed during awake surgery for right-hemispheric gliomas.Indications for utilizing awake surgery in the right cerebral hemisphere.Evaluation methods and tools for intraoperative functional assessment.

Quality assessment: the selected papers underwent a quality assessment to ensure the reliability and validity of the information provided. This involved evaluating the study design, methodology, and the clarity of the reported results.

Synthesis of results: a synthesis of the findings from the selected papers was conducted to provide a comprehensive overview of the current state of knowledge regarding awake surgery in right-hemispheric lesions, with a focus on anatomical considerations, surgical techniques, and intraoperative functional evaluation methods.

## 3. Results

Our comprehensive literature search yielded a total of 74 sources (Figure 1), comprising original articles, books, monographs, and review articles. However, only two original papers provided detailed insights into a large series of language assessment cases. These studies collectively included 246 patients who underwent “surgery with awakening and intraoperative electrostimulation mapping”, providing extensive information on neurological semiology and mapping techniques. It is noteworthy that most of the remaining studies primarily focused on neuroradiological and neuroimaging aspects.

### 3.1. Understanding Syndromes and Symptoms of Right-Hemispheric Lesions and Their Intraoperative Tests

Objective neuropsychological studies have reported that cognitive and behavioral deficits after brain surgery are often evident even in the right hemisphere [40]. Therefore, for an optimal quality of life, particularly in patients with extended survival, such as those with low-grade gliomas, the resection of right-sided tumors should be performed using awake surgery with cortical and axonal electrostimulation mapping [41,42]. The central roles of the right hemisphere encompass motor execution and control, visual processes, spatial cognition, verbal and nonverbal semantic processing, executive functions (such as attention), and social cognition (mentalization and affect). In this section, we will review the fundamental clinical symptoms and signs that develop in lesions of right hemisphere tumors during intraoperative direct electrical stimulations, along with the corresponding intraoperative testing methods.

#### 3.1.1. Lesions on the Frontal Lobe near the Superior Frontal Gyrus Involving the Supplementary Motor Area (SMA): Assessment of Supplementary Motor Functions

When a lesion is located on or near the pre- or postcentral gyri, motor-evoked potential (MEP) and somatosensory-evoked potential (SSEP) monitoring become mandatory, offering real-time information to the neurosurgeon about the state of the primary motor area and corticospinal and corticonuclear tracts. However, it is equally crucial to monitor supplementary motor functions. Axonal stimulation of the SMA, frontal aslant tract (FAT), and fronto-striatal tracts in awake patients performing continuous movements can result in disturbances of motor initiation and control, ranging from complete arrest to an involuntary acceleration of movement. While postoperative SMA syndrome is generally considered transient, Briggs et al. (2021) reported that neglecting the preservation of FAT fibers (originating from the supplementary motor area) can lead to permanent deficits in 13% of patients [43]. Furthermore, unilateral subcortical direct electrical stimulation of the right hemisphere can not only disrupt left movement but also affect the movement of both hands during a bimanual coordination task [44].

A higher microstructural organization of the bilateral FAT is associated with lower acceleration and deceleration amplitudes for reach and reach-to-grasp movements, indicating more efficient visuomotor processing and leading to smoother movement trajectories. Testing involves instructing patients intraoperatively to move both hands in parallel to speech (dual task), assessing the possible occurrence of acceleration or complete arrest (hemicorporeal akinesia) indicative of SMA syndrome (Table 1).

#### 3.1.2. Deep Frontal Lobe Lesions Involving the FAT: Assessment of Executive Functions

Several studies have emphasized the role of the right FAT in executive functions. Executive functions, encompassing inhibition, working memory, planning, and monitoring, are often associated with the frontal lobe, and their impairment can have significant negative implications for an individual’s social and professional life [45,46]. The right hemisphere’s modulatory role in the cortico–subcortical network of higher mental functions has been supported by findings, with fibers anterior to the corticospinal tract originating from the supplementary motor area, lateral premotor cortex, and the depth of the precentral sulcus, forming the fronto-striatal tract [47].

Despite the importance of the right frontal lobe for cognitive and emotional functions, intraoperative monitoring of these functions has been rare due to the complex nature of emotional functions and the challenge of adapting standard bedside tasks to awake surgery conditions. However, dysexecutive disorders can significantly impact a patient’s quality of life by deteriorating abstract reasoning, judgment abilities, and reducing mental flexibility. These behavioral disturbances, particularly in social life, can manifest as social inappropriateness, extreme ebullience, and aggression [48]. In some cases, lesions involving the right frontal lobe may initially manifest mildly and be less obvious than language and motor disabilities. Moreover, the intraoperative assessment of corresponding functions poses technical challenges, and recovery from symptoms depends on the extent of injured fibers. Testing executive functions intraoperatively is particularly challenging due to their complicated nature, with only a minority of the published reports available in the current literature. Some authors have adapted the Stroop test for awake brain surgery, utilizing it for intraoperative monitoring of frontal functions. The Stroop test assesses the functioning of the anterior cingulate cortex, which is crucial for managing conflicts. The presence of the Stroop effect can be evaluated as a measure of the anterior cingulate cortex’s functioning. This test offers the advantage of high specificity for use in awake surgery as it can assess three basic functions of the anterior cingulate cortex: intention to action (dysfunction causes akinetic mutism), motor initiation, and inhibitory function (suppressing inappropriate responses). Other testing methods such as the Wisconsin Card Sorting Test, Trail Making Test, Nelson Modified Test, and Spatial Memory Test require at least 20 min for evaluation and are not suitable for intraoperative conditions. The common characteristics of these tests, demanding more time to plan and involving hand movements for drawing, make them time-consuming, potentially explaining their limited use in intraoperative settings.

#### 3.1.3. Lesions of the Medial Part of the Frontal Lobe Involving the Cingulate Gyrus: Assessment of Social Cognition

Social cognition encompasses theory of mind, nonverbal language—primarily facial emotion recognition—and empathy. Despite the lack of consensus on the anatomical demarcations of social cognition function, several studies have indicated that these functions are distributed across various regions of the “facial network,” including the anterior cingulate gyrus, the medial ventral PFC, the gyrus rectus, the medial aspect of the superior frontal gyrus, both occipito-temporal cortices, and the posterior part of the right superior temporal sulcus (STS) (Figure 2). Consequently, any lesion infiltrating these areas may lead to disorders in one or several functions of social cognition. Transcranial magnetic stimulation of the posterior part of the right STS has been shown to improve the recognition of emotions in healthy volunteers [49]. The cingulum, which links the rostral medial prefrontal cortex/anterior cingulate and the medial posterior parietal cortex (including the posterior cingulate cortex and ventral precuneus), is involved in the default mode network and may participate in some aspects of conscious information processing. The disruption of this subcortical connectivity on the posterior cingulate cortex can lead to a breakdown in consciousness in awake patients, resulting in transient behavioral unresponsiveness and a loss of connectedness to the external environment [50]. Another study supported the role of the right hemisphere in emotional and behavioral disorders in patients with frontotemporal lobar degeneration (FTD), positing a dominant role of the right hemisphere for emotional functions [51,52,53,54,55]. There are also reports about the role of the amygdala and insula in facial emotion recognition. The scattered nature of social cognition functions suggests using corresponding testing methods in cases where any of these centers are affected by lesions.

It is noteworthy that emotion recognition was also tested by the left insular lobe in a study involving 13 patients, where it was found that there was an insignificant decrease in emotion recognition [49]. Testing for facial emotion recognition can be conducted using Ekman’s six primal facial emotions (anger, happiness, fear, surprise, disgust, and sadness), with a testing time of 10 s per image. Other testing methods, such as “Reading the Mind in the Eyes”, may be more challenging in intraoperative settings due to the observation of only the eyes. Additionally, methods like the JACMAN (Japan, Caucasian brief affect recognition test), involving 56 items measuring different aspects of expressions and multiple items representing each aspect, are time-consuming and not intraoperatively applicable.

#### 3.1.4. Lesions of the Temporoparietal Region: Assessment of Visuospatial Function

We have not identified a standard anatomically defined area for the cortical center of visuospatial function. Instead, several cortical regions and subcortical tracts have been suggested as responsible areas for visuospatial cognition, validated through direct electrical stimulation. These areas include the ventral frontal cortex (VFC), corresponding generally to the middle frontal gyrus (MFG) and inferior frontal gyrus (IFG), the temporoparietal junction where the inferior parietal lobule (comprising two gyri—supramarginal and angular) meets the superior temporal gyrus, and finally, the insula. The manifestation of irritation or dysfunction in these regions will ultimately result in unilateral neglect (UN).

#### 3.1.5. Lesions Localized Deep in the Basal Surface of the Right Occipito-Temporal Area, Involving ILF, IFOF, and SLF: Assessment of Visuospatial Cognition

Studies on the direct electrical stimulation of the right optic radiation in awake patients have shown that beyond the primary visual cortex, stimulation can induce inhibitory phenomena such as blurred vision or the impression of a shadow, as well as “excitatory phenomena” manifested as visual hallucinations and metamorphopsias. Stimulation of the right inferior longitudinal fasciculus (ILF) may lead to left visual hemiagnosia, resulting from the disruption of occipital visual input and the fusiform gyrus [56]. These data support the key role of the inferior longitudinal fasciculus in visual recognition in the right hemisphere. Additionally, stimulation of the SLF II and supramarginal gyrus can cause disturbances in spatial cognition and a rightward deviation in line bisection tests. Cortical centers in these right perisylvian regions provide information about the position and motion of our body in space, playing a critical role in regulating body position in relation to external space [57,58,59,60]. This perisylvian neural network is vital for the neural transformation of converging vestibular, auditory proprioceptive, and visual inputs into spatial representations [49]. These findings underscore the pivotal role of the right hemisphere fronto-parietal network in spatial awareness and visual scene processing. Some studies have even suggested a role of the right SLF in vestibular syndrome inducing vertigo, indicating its contribution to body posture and spatially oriented actions [47].

Testing: various assessment tests for hemispatial neglect have been described in the literature. The Catherine Bergego Scale, comprising 10 everyday tasks observed by the doctor during self-care activities, is one example [61,62]. However, it is not suitable for intraoperative settings. Other tests, such as clock face drawing and butterfly drawing tests, may be time-consuming and inconvenient for patients in a lying position.

Intraoperative testing: two tests are particularly well-suited for intraoperative settings due to their reliability and feasibility, taking less than 5 min and being easy in nature:

Line bisection test (Alberts test): in this test, the patient marks the center of a given horizontal line. Rightward displacement of the bisection indicates irritation of the cortical or subcortical regions of spatial cognition. A lateral deviation of the patient’s mark from the midline of approximately 5 mm is considered a positive result.

Target cancellation test: patients are presented with a collection of simple, distinct shapes (a star, triangle, square, and circle) randomly arranged with a bisecting vertical line dividing the array into equal halves. Patients are instructed to mark or “cancel” the specified targets or objects in both the right and left fields. Note that these tests may not be suitable for patients with hemianopia, as they may yield false positive results.

#### 3.1.6. Lesions Localized in Temporal Lobe and Insula: Assessing Emotional Prosody

Emotional prosody, a fundamental aspect of social cognition, is associated with cortical centers primarily located in the superior temporal gyrus and partially in the inferior frontal gyrus. However, some studies have also reported cortical activation sites in the right supramarginal gyrus (SMG) with the middle and superior temporal gyri with functional magnetic resonance imaging (fMRI) [49]. It is crucial to note that linguistic prosody is processed in both hemispheres, indicating that its function is not solely anatomically confined to the right side. The impact of the right ventral pathway on non-verbal semantics has been explored in numerous studies. Patients with low-grade gliomas (LGGs) involving ventral tracts underwent awake craniotomy, revealing disturbances in non-verbal semantic functions during tests. Semantic impairments, such as semantic verbal paraphasia during naming tasks at the level of the pars triangularis, dorsolateral prefrontal cortex, and opercular cortex, were observed [63,64]. Similar findings were confirmed at the subcortical level, inducing verbal semantic impairment in the right inferior fronto-occipital fasciculus (IFOF), emphasizing the critical role of the right ventral stream in these processes [65]. The right superior longitudinal fasciculus (SLF) has also been investigated for its crucial role in simultaneous enrollment of both verbal and non-verbal functions—language and spatial cognition—during the performance of dual tasks (limb movement and object naming) in awake craniotomy with direct electrical stimulation [66]. Direct electrical stimulation helps elicit symptoms by triggering cortical centers and subcortical tracts. However, these symptoms will not persist if the corresponding areas are preserved. Studies following aggressive resection of parietal lobe gliomas without subcortical mapping have reported that 13% of patients experienced postoperative dysphasia [10]. Fan et al. found that the right anterior insula was linked with the affective–perceptual form of empathy, while the left insula was associated with both the affective–perceptual and cognitive–evaluative forms of empathy. The role of the insula in empathy and social cognition has been confirmed in lesion studies [2]. Testing: emotional prosody is assessed through tests where patients are asked to read different sentences with specific emotional tones. The performance in these tests is subjectively analyzed by an expert physician to determine whether patients exhibit aprosodic or normal prosody. The complexity of these tests and their subjective interpretations explain the challenges in investigating prosody during awake surgery, contributing to the absence of a standardized procedure.

**Table 1 cancers-16-01161-t001:** Right-hemispheric functions and correlated neurological deficits, their testing methods, and surgical considerations.

	Functional Area	Gyrus	Responsible Subcortical Tract	Function	Deficit	Testing Methods	Awake Intraoperative Assessment
1	Ventral frontal cortex (vPFC)	MFG IFG	SLF III SLF II IFOF	Visuospatial cognition	Somatoparaphrenia Anosognosia Unilateral neglect Allochiria	Line bisection test Alberts test [67] Catherine Bergego Scale (CBS) [62] Target cancellation test Clock face drawing Butterfly drawing	Line bisection test Target cancellation test
2	Temporoparietal junction (TPJ)	SMG, both MTG STG (posterior part)	IFOF Right UF + corticolimbic system with bilateral mvPFC and orbitofrontal cortex and precuneus	Visuospatial Social cognition empathy, TOM	Somatoparaphrenia Anosognosia Unilateral neglect Constructional apraxia Emotional dysprosody Undermentalizing (autistic) Overmentalizing (schizophrenic) accentuation	Line bisection test Alberts test CBS Benton Visual Retention Test [68] “Reading the mind in the eyes” test [53]. False belief vs. photo	Line bisection test Target cancellation test
3	Medial ventral prefrontal cortex (mvPFC)	Ant.Cingulate gyrus Gyrus rectus Medial SFG		Social cognition (empathy TOM)	“Mind blindness” Undermentalizing (autistic) Overmentalizing (schizophrenic) accentuation	“Reading the mind in the eyes” test. Strategic game Trait judgement Social animations Rational actions	Reading the Mind in the Eyes
4	Facial network, temporal part. SMG, left insular area.	Bilateral FFG posterior STS IFG, orbitofrontal gyrus mvPFC61 anterior cingulate gyrus, gyrus rectus medial SFG	UF	Facial emotion recognition		Ekman’s face test Japan, Caucasian brief affect recognition test	Ekman’s faces
5	Emotional prosody	IFG SMG rt STG	AF	Emotional prosody	Emotional dysprosody	Storytelling with an intonation or listening to a prosodic text with emotional background.	No intraoperative test documented in literature
6	Empathy	Bilateral mvPFC Bilateral TPJ STS paracingulate IFG, cingulate gyrus, and amygdala	UF	Showing empathy		Reading the Mind in the Eyes False belief vs. photograph Strategic game Trait judgement Social animations Rational actions	Reading the Mind in the Eyes Balanced Emotional Empathy Scale (BEES)
7	Theory of mind	Bilateral mvPFC Bilateral TPJ STS Lat orbitofrontal gyrus MFG, cuneus precuneus, and STG	Corticolimbic system	Theory of mind	“Mind blindness” Undermentalizing (autistic) [53,55] Overmentalizing (schizophrenic) accentuation		

## 4. Discussion

A study involving 658 consecutive cases [69] of intraoperative somatosensory-evoked potential (SEP) monitoring conducted by authors from Essen, Germany, demonstrated that the sensitivity of neurophysiological intraoperative monitoring (IOM) is typically around 80% [1]. The primary objective of IOM is to promptly identify patients at risk of neurological impairment during surgery, enabling the surgical team to react promptly. This underscores the limitation of relying solely on IOM under general anesthesia for the optimal preservation of complex motor functions. Studies utilizing the direct electrical stimulation of the right hemisphere by various authors have revealed significant interindividual structural and functional variabilities, particularly at the cortical level [2]. This variability is attributed to the complex networking of white matter tracts, emphasizing that no single function is mediated by a single cortical area [70,71]. Instead, it results from the interactions of large-scale subcircuits, allowing for neural reshaping over time to compensate for injury [72].

### 4.1. Direct Electrical Stimulation (DES)

Direct electrical stimulation (DES) is a pivotal technique used in neurosurgery for the functional mapping of eloquent cortical areas and subcortical white matter tracts [12]. By delivering low-amplitude electrical currents directly to specific brain regions, DES allows neurosurgeons to assess functional integrity and determine the localization of critical structures during awake craniotomy procedures.

#### 4.1.1. Stimulation Parameters

During DES, electrical currents are typically delivered using specialized electrodes, such as bipolar or monopolar probes, which are inserted into the brain tissue under direct visualization. The stimulation parameters, including frequency, duration, and intensity of electrical pulses, are carefully adjusted based on individual patient factors and the specific functional regions being targeted. Common stimulation frequencies range from 50 to 60 Hz, with pulse durations of 1 to 5 milliseconds and intensities typically ranging from 1 to 10 mA.

#### 4.1.2. Site Selection Criteria

The selection of stimulation sites is guided by preoperative neuroimaging data, including functional MRI (fMRI) and diffusion tensor imaging (DTI), which provide information about the spatial relationship between the tumor and eloquent brain regions. Additionally, intraoperative electrocorticography (ECoG) may be used to identify areas of abnormal electrical activity indicative of functional significance. Stimulation sites are strategically chosen to encompass language, motor, and cognitive areas potentially at-risk during tumor resection, with particular attention to regions proximal to the lesion and critical white matter pathways.

#### 4.1.3. Protocols for Patient Safety

Ensuring patient safety during DES procedures is paramount. Neurosurgeons adhere to strict protocols to minimize the risk of adverse events, including seizures and neurological deficits. Prior to stimulation, patients are typically administered antiepileptic medications to reduce the likelihood of seizure activity. Additionally, continuous intraoperative monitoring of neurological function, including speech and motor responses, allows for real-time assessment of the effects of electrical stimulation on brain function. Stimulation is incrementally delivered, starting at low intensities, and gradually increasing to threshold levels while closely monitoring for any signs of functional impairment. If a significant functional response is elicited, such as speech arrest or motor weakness, the stimulation is immediately ceased to prevent permanent neurological deficits.

Research on neuroplasticity has highlighted that functions cannot be reliably localized solely based on anatomical criteria [4]. The individual organization of the cerebral cortex varies among individuals, emphasizing the importance of individualized mapping for safer resection with minimal neurological consequences. Positive mapping aids in better surgical approach selection and the delineation of lesion resection limits, crucial for avoiding permanent postoperative neurological deficits [5,6,7].

The involvement of the right hemisphere becomes evident in cases of injury or post-operative damage to eloquent zones of the left hemisphere. For instance, resection of the left supplementary motor area (SMA) can lead to SMA syndrome, with patient recovery demonstrated by functional magnetic resonance imaging (fMRI) revealing compensation by recruitment of the right-sided SMA and ipsilateral primary motor cortex [8]. Reports have also indicated that the resection of insular low-grade gliomas (LGGs) involving deep gray matter nuclei may not cause cognitive disorders due to compensation through recruitment of parallel subcortical pathways. High-order cognitive processes are sustained by large-scale networks distributed throughout the brain, challenging the concept of specific functions attributed to cortical regions. Focal cortical electric stimulation demonstrates specific dysfunction but may not fully reveal the functioning of subcortical circuits. Localizationism can be misleading for neurosurgeons, particularly in cases of higher-order cognitive functions, emphasizing the crucial need for intraoperative cortical/subcortical mapping.

Despite the importance of mapping right-hemispheric functions, there is currently no standardized test battery for assessing these functions in intraoperative settings. Developing more accurate and easily applicable neuropsychological tests for mapping right-hemispheric lesions is essential. The challenge lies in the multidisciplinary team’s responsibility, including neurosurgeons, neuropsychologists, psychiatrists, and neurologists, to address the demands of intraoperative settings, considering the time-consuming nature of available neurocognitive tests.

### 4.2. Limitations of Current Studies and Future Directions

While existing literature has provided valuable insights into neurosurgical mapping techniques and their clinical applications, it is essential to acknowledge several limitations inherent in the current body of research. These limitations contribute to gaps in our understanding and highlight areas for further investigation to advance the field. One notable limitation is the variability in study methodologies, including differences in patient populations, lesion characteristics, mapping techniques, and outcome measures [9]. Such heterogeneity makes it challenging to compare findings across studies and draw definitive conclusions regarding the efficacy and generalizability of specific mapping approaches [73]. Moreover, many studies examining the utility of neurosurgical mapping techniques were limited by relatively small sample sizes, which may have affected the statistical power and robustness of their findings. Larger, multicenter studies with standardized protocols are needed to validate the effectiveness of these techniques across diverse patient populations and clinical settings. Additionally, while awake craniotomy has emerged as a valuable tool for preserving language and cognitive functions during brain surgery, its applicability may be limited by patient factors such as anxiety, cooperation, and tolerance of the procedure. Future research should explore strategies to optimize patient selection, enhance intraoperative monitoring, and mitigate potential adverse effects to maximize the benefits of awake surgery while minimizing patient discomfort. Furthermore, the current literature predominantly focuses on mapping techniques for language and motor functions, with less emphasis on cognitive domains such as memory, attention, and executive function. Future studies should investigate the feasibility and efficacy of incorporating comprehensive cognitive mapping protocols into neurosurgical practice to minimize postoperative cognitive deficits and improve overall patient outcomes. Emerging technologies, such as functional neuroimaging modalities (e.g., functional MRI, diffusion tensor imaging, etc.) and intraoperative neurophysiological monitoring techniques, offer promising avenues for advancing our understanding of brain function and refining surgical mapping approaches [10,11]. Continued research in these areas may lead to the development of more precise and individualized mapping strategies that optimize functional preservation while maximizing tumor resection. In conclusion, while neurosurgical mapping techniques have revolutionized the management of brain lesions, there remain important challenges and opportunities for further research. Addressing the limitations of current studies and exploring novel methodologies and technologies will be crucial for advancing the field and improving patient outcomes in neurosurgical practice.

## 5. Conclusions

In summary, this review of intraoperative neurophysiological studies underscores the critical role of awake craniotomy with intraoperative electrostimulation mapping in gliomas affecting the non-dominant right hemisphere of the brain. The term “non-dominant” may not be entirely suitable for the right hemisphere, given its significant contribution to individual and social life by providing fundamental and irreplaceable cognitive functions [74,75]. The loss of these functions can lead to a substantial deterioration in quality of life. While there are ample reports on intraoperative language and motor functions, the assessment of executive cognitive functions such as memory, calculation, emotions, and working memory has been reported in limited quantity. This underscores a notable underexposure of executive functions controlled by the right hemisphere. There is a clear need to shift the paradigm and evolve assessment methodologies in this direction. Based on the analysis of the current literature review, it is advisable to incorporate intraoperative mapping, particularly in patients with little or no preoperative neurological deficits. This approach can enhance the understanding and preservation of the intricate cognitive functions associated with the right hemisphere, ultimately contributing to improved surgical outcomes and postoperative quality of life.

## Figures and Tables

**Figure 1 cancers-16-01161-f001:**
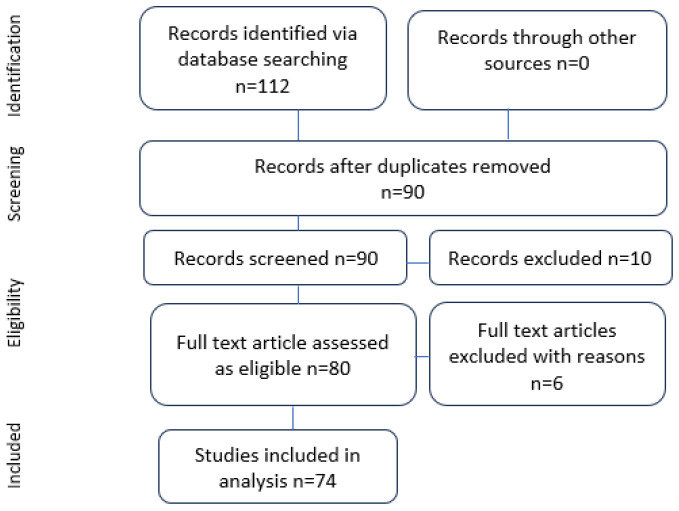
Preferred Reporting Items for Systematic Reviews and Meta-Analyses (PRISMA) flowchart of the study.

**Figure 2 cancers-16-01161-f002:**
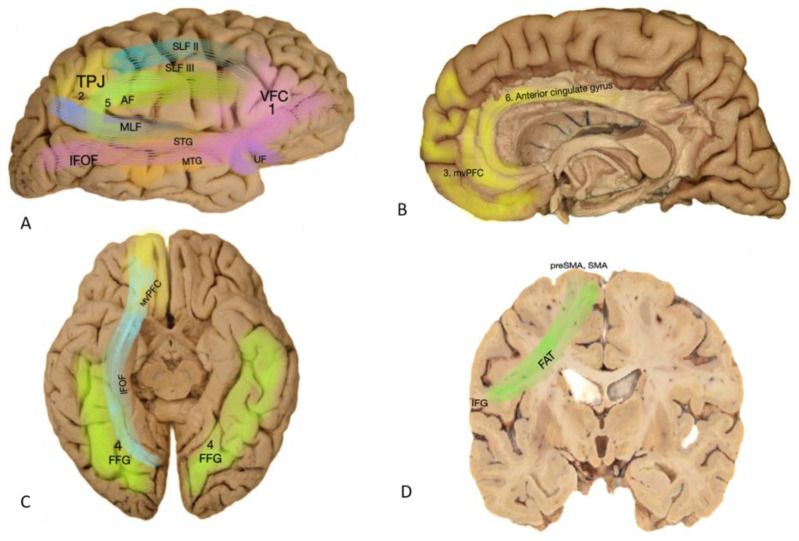
Basic functional–eloquent regions of the right hemisphere. (**A**) Lateral view of the right hemisphere. VFC (ventral frontal cortex) in pink and TPJ (temporoparietal junction) in yellow, and interconnecting subcortical pathways are depicted. (**B**) Medial view of the right hemisphere. Middle ventral prefrontal cortex (mvPFC) and anterior cingulate gyrus can be observed. (**C**) Bottom image of the bilateral hemisphere. Basal view of the mvPFC and bilateral fusiform gyrus (FFG), which is responsible for the facial network. There is also a deep layer of IFOF illustrated connecting the fusiform gyrus on its way to anterior temporal regions and reaching up to the middle ventral prefrontal cortex. (**D**) Coronal section of the posterior frontal lobe. FAT: frontal aslant tract, which is a subcortical tract connecting the preSMA and IFG, is illustrated.

## Data Availability

The data can be shared up on request.

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
