# Peer review of "Awake Craniotomy for Gliomas in the Non-Dominant Right Hemisphere: A Comprehensive Review"

_cancers, 2024, doi:10.3390/cancers16061161_

Round 1
Reviewer 1 Report
Comments and Suggestions for Authors
Authors point out the importance of intraoperative neurological monitoring of higher cognitive functions during glioma surgery. The Review is well written and discussing relevant issue, which is surely extremely important for further improvement of functional outcome an QoL. Nevertheless, it would be interesting to identify more clearly characteristics of patients who could basic ally underwent resection under general anesthesia with acceptable risk but I understand that this was not the main question.
Author Response
Reviewer 1:
Authors point out the importance of intraoperative neurological monitoring of higher cognitive functions during glioma surgery. The Review is well written and discussing relevant issue, which is surely extremely important for further improvement of functional outcome an QoL. Nevertheless, it would be interesting to identify more clearly characteristics of patients who could basic ally underwent resection under general anesthesia with acceptable risk, but I understand that this was not the main question.
Our response to Reviewer 1:
Thank you for your feedback. We appreciate your recognition of the importance of intraoperative neurological monitoring of higher cognitive functions during glioma surgery as discussed in our review. While we did not specifically address the characteristics of patients who could undergo resection under general anesthesia with acceptable risk, we acknowledge that this could be an interesting avenue for further investigation. Understanding the patient characteristics that might make them suitable candidates for resection under general anesthesia could indeed contribute to optimizing surgical outcomes and quality of life. We will consider incorporating this aspect into future research.
Reviewer 2 Report
Comments and Suggestions for Authors
Thank you for the opportunity to review this article, which I read with interest. Since the seminal work of Penfield, great advance has been achieved in monitoring neurological functions during brain surgery, mostly for preserving language and motor function. Nowadays, better anesthetic control allows surgeons to perform awake surgeries to localize and adapt the surgical resection to avoid damaging crucial neural tracts, while keeping patient comfortable enough to collaborate finely.
As the authors point, awake surgeries are mostly indicated for left hemisphere lesions, but as neuroscience knowledge grows and patients are more carefully monitored, we are now aware that right hemisphere lesions can be responsible of serious psychological and neurological sequaelae that can negatively affect patient's life.
It is therefore pertinent to present this brief review that enlightens the need to adopt awake surgery for a broader scope of left and right brain lesions as the gold standard for function preservation. This is true for low grade gliomas, where neuroplasticity will help in case of following need for new resection of residual tumor. But it is also very important for high grade gliomas, where functional integrity is of great importance for the patient to receive complementary treatments.
This comprehensive review not only serves to understand that the right hemisphere shouldn't be considered non dominant, because of the potential damage that "blind" surgery can inflict to the patient but also because both hemispheres are interconnected and many unexpected contralateral functions may be harmed.
The authors also expose the need for accurate and easily applicable tests for surgical mapping of right hemisphere in awake surgery, offering new multidisciplinary investigational venues.
They conclude, and I agree with this vision, that monitoring and mapping right hemisphere in awake patines will enhance the understanding of its function and help improve surgical results for our patients.
Author Response
Reviewer 2:
Thank you for the opportunity to review this article, which I read with interest. Since the seminal work of Penfield, great advance has been achieved in monitoring neurological functions during brain surgery, mostly for preserving language and motor function. Nowadays, better anesthetic control allows surgeons to perform awake surgeries to localize and adapt the surgical resection to avoid damaging crucial neural tracts, while keeping patient comfortable enough to collaborate finely.
As the authors point, awake surgeries are mostly indicated for left hemisphere lesions, but as neuroscience knowledge grows and patients are more carefully monitored, we are now aware that right hemisphere lesions can be responsible of serious psychological and neurological sequalae that can negatively affect patient's life.
It is therefore pertinent to present this brief review that enlightens the need to adopt awake surgery for a broader scope of left and right brain lesions as the gold standard for function preservation. This is true for low grade gliomas, where neuroplasticity will help in case of following need for new resection of residual tumor. But it is also very important for high grade gliomas, where functional integrity is of great importance for the patient to receive complementary treatments.
This comprehensive review not only serves to understand that the right hemisphere shouldn't be considered non dominant, because of the potential damage that "blind" surgery can inflict to the patient but also because both hemispheres are interconnected, and many unexpected contralateral functions may be harmed.
The authors also expose the need for accurate and easily applicable tests for surgical mapping of right hemisphere in awake surgery, offering new multidisciplinary investigational venues.
They conclude, and I agree with this vision, that monitoring and mapping right hemisphere in awake patients will enhance the understanding of its function and help improve surgical results for our patients.
Our response to Reviewer 2:
Thank you for your thorough review and insightful comments on our article. We appreciate your recognition of the importance of advancing neurological monitoring techniques during brain surgery, particularly in the context of awake surgeries for both left and right hemisphere lesions. You've accurately highlighted the evolving understanding of right hemisphere functions and the potential negative impacts of "blind" surgery on patients' lives, emphasizing the interconnectedness of both hemispheres and the importance of preserving functional integrity. Furthermore, your points regarding the significance of awake surgery for low and high-grade gliomas, as well as the need for accurate mapping techniques for the right hemisphere, are well-taken. We agree with your assessment that adopting awake surgery as the gold standard for function preservation in a broader scope of brain lesions is essential for optimizing surgical outcomes and improving patients' quality of life. We also acknowledge the need for further multidisciplinary investigation into developing accurate and easily applicable tests for surgical mapping of the right hemisphere.
Thank you for your valuable feedback, which reinforces the importance of our work in advancing the understanding and practice of neurological monitoring during glioma surgery.
Reviewer 3 Report
Comments and Suggestions for Authors
The manuscript presents a comprehensive review of the advanced mapping techniques used in neurosurgery for identifying language and cognitive functions. It offers a detailed examination of the neuroanatomical substrates involved in language and cognitive processing, the clinical implications of lesions in these areas, and the methodologies for functional mapping during neurosurgical procedures. The review integrates findings from various studies, providing a broad perspective on the subject matter. However, while the manuscript is informative and well-structured, there are areas where improvements can enhance its clarity, accuracy, and contribution to the field.
Major Comments
1. Clarity and Organization: The manuscript is generally well-organized, with sections logically structured to guide the reader through the complexities of neuroanatomical and functional relationships. However, the transitions between sections could be smoother to enhance the narrative flow. Additionally, some sections are dense with technical jargon, which could be simplified or explained in lay terms to make the review more accessible to readers outside the immediate field of neurosurgery or neurology.
Issue Example: The transition between the sections discussing the anatomical basis of cognitive functions and the review of surgical mapping techniques is abrupt. Readers might find it challenging to see how these sections are interconnected.
Suggested Action: Include a bridging paragraph that clearly explains how the anatomical understanding of cognitive functions underpins the development and application of surgical mapping techniques. This would smooth the transition and enhance the manuscript's narrative flow.
2. Critical Analysis and Discussion: While the manuscript does an excellent job of summarizing existing literature, it could benefit from a more critical analysis of the studies discussed. For example, highlighting the limitations of current methodologies or discrepancies in findings across studies would provide a more nuanced understanding of the state of the field. Additionally, a discussion on the future directions for research and clinical practice, including emerging technologies and methodologies, would be valuable.
Issue Example: The manuscript summarizes findings from various studies on the efficacy of awake craniotomy in preserving cognitive functions without critically discussing conflicting evidence or the limitations of the studies mentioned.
Suggested Action: Incorporate a subsection that critically evaluates the limitations of current studies, such as sample size, methodology biases, or the generalizability of findings. Discuss how these limitations could impact the interpretation of results and suggest areas for future research to address these gaps.
3. Methodological Details: The manuscript mentions various mapping techniques and tests used in neurosurgery but does not consistently provide sufficient details on the methodologies involved. Expanding on how these techniques are applied in clinical settings, their accuracy, and potential limitations would significantly enhance the manuscript's utility for readers interested in the practical aspects of neurosurgical mapping.
Issue Example: The description of direct electrical stimulation (DES) techniques is somewhat vague, lacking details on the specific protocols used (e.g., stimulation frequencies, durations, and thresholds for mapping).
Suggested Action: Expand the section on DES to include technical specifics, such as typical stimulation parameters and how adjustments are made based on patient responses. Also, discuss the criteria for selecting mapping sites and the protocols for ensuring patient safety during these procedures.
Minor Comments:
Consistency in Terminology: Ensure consistency in the use of technical terms and abbreviations throughout the manuscript. Defining abbreviations at their first occurrence and maintaining consistent terminology would aid in readability and comprehension.
Issue Example: The term "awake surgery" is used interchangeably with "awake craniotomy" throughout the document. While these terms are closely related, their usage interchangeably without clear definitions might confuse readers unfamiliar with the nuances of neurosurgical procedures.
Suggested Action: Define terms clearly when they first appear in the text and maintain consistent use throughout the manuscript. For example, you might choose to use "awake craniotomy" consistently and note that it is a type of awake surgery specific to procedures involving the cranium.
Comments on the Quality of English Language1. Clarity and Organization: The manuscript is generally well-organized, with sections logically structured to guide the reader through the complexities of neuroanatomical and functional relationships. However, the transitions between sections could be smoother to enhance the narrative flow. Additionally, some sections are dense with technical jargon, which could be simplified or explained in lay terms to make the review more accessible to readers outside the immediate field of neurosurgery or neurology.
2. Consistency in Terminology: Ensure consistency in the use of technical terms and abbreviations throughout the manuscript. Defining abbreviations at their first occurrence and maintaining consistent terminology would aid in readability and comprehension.
Author Response
Reviewer 3:
The manuscript presents a comprehensive review of the advanced mapping techniques used in neurosurgery for identifying language and cognitive functions. It offers a detailed examination of the neuroanatomical substrates involved in language and cognitive processing, the clinical implications of lesions in these areas, and the methodologies for functional mapping during neurosurgical procedures. The review integrates findings from various studies, providing a broad perspective on the subject matter. However, while the manuscript is informative and well-structured, there are areas where improvements can enhance its clarity, accuracy, and contribution to the field.
Major Comments
- Clarity and Organization: The manuscript is generally well-organized, with sections logically structured to guide the reader through the complexities of neuroanatomical and functional relationships. However, the transitions between sections could be smoother to enhance the narrative flow. Additionally, some sections are dense with technical jargon, which could be simplified or explained in lay terms to make the review more accessible to readers outside the immediate field of neurosurgery or neurology.
Issue Example: The transition between the sections discussing the anatomical basis of cognitive functions and the review of surgical mapping techniques is abrupt. Readers might find it challenging to see how these sections are interconnected.
Suggested Action: Include a bridging paragraph that clearly explains how the anatomical understanding of cognitive functions underpins the development and application of surgical mapping techniques. This would smooth the transition and enhance the manuscript's narrative flow.
- Critical Analysis and Discussion: While the manuscript does an excellent job of summarizing existing literature, it could benefit from a more critical analysis of the studies discussed. For example, highlighting the limitations of current methodologies or discrepancies in findings across studies would provide a more nuanced understanding of the state of the field. Additionally, a discussion on the future directions for research and clinical practice, including emerging technologies and methodologies, would be valuable.
Issue Example: The manuscript summarizes findings from various studies on the efficacy of awake craniotomy in preserving cognitive functions without critically discussing conflicting evidence or the limitations of the studies mentioned.
Suggested Action: Incorporate a subsection that critically evaluates the limitations of current studies, such as sample size, methodology biases, or the generalizability of findings. Discuss how these limitations could impact the interpretation of results and suggest areas for future research to address these gaps.
- Methodological Details: The manuscript mentions various mapping techniques and tests used in neurosurgery but does not consistently provide sufficient details on the methodologies involved. Expanding on how these techniques are applied in clinical settings, their accuracy, and potential limitations would significantly enhance the manuscript's utility for readers interested in the practical aspects of neurosurgical mapping.
Issue Example: The description of direct electrical stimulation (DES) techniques is somewhat vague, lacking details on the specific protocols used (e.g., stimulation frequencies, durations, and thresholds for mapping).
Suggested Action: Expand the section on DES to include technical specifics, such as typical stimulation parameters and how adjustments are made based on patient responses. Also, discuss the criteria for selecting mapping sites and the protocols for ensuring patient safety during these procedures.
Minor Comments:
Consistency in Terminology: Ensure consistency in the use of technical terms and abbreviations throughout the manuscript. Defining abbreviations at their first occurrence and maintaining consistent terminology would aid in readability and comprehension.
Issue Example: The term "awake surgery" is used interchangeably with "awake craniotomy" throughout the document. While these terms are closely related, their usage interchangeably without clear definitions might confuse readers unfamiliar with the nuances of neurosurgical procedures.
Suggested Action: Define terms clearly when they first appear in the text and maintain consistent use throughout the manuscript. For example, you might choose to use "awake craniotomy" consistently and note that it is a type of awake surgery specific to procedures involving the cranium.
Comments on the Quality of English Language
- Clarity and Organization: The manuscript is generally well-organized, with sections logically structured to guide the reader through the complexities of neuroanatomical and functional relationships. However, the transitions between sections could be smoother to enhance the narrative flow. Additionally, some sections are dense with technical jargon, which could be simplified or explained in lay terms to make the review more accessible to readers outside the immediate field of neurosurgery or neurology.
- Consistency in Terminology: Ensure consistency in the use of technical terms and abbreviations throughout the manuscript. Defining abbreviations at their first occurrence and maintaining consistent terminology would aid in readability and comprehension.
Our response to Reviewer 3:
Thank you for your thorough and constructive feedback on our manuscript. We appreciate the detailed insights provided, and we are committed to addressing the issues you raised to enhance the clarity, accuracy, and contribution of our review to the field of neurosurgery. Below are our responses to each of your major and minor comments:
Major Comments:
Clarity and Organization:
Issue Example: We acknowledge the abrupt transition between sections discussing the anatomical basis of cognitive functions and surgical mapping techniques. We included a bridging paragraph to explain how the anatomical understanding informs the development and application of surgical mapping techniques, thereby improving the narrative flow.
Critical Analysis and Discussion:
Issue Example: We understand the need for a more critical analysis of the studies discussed, including their limitations and discrepancies. We incorporated a subsection to evaluate the limitations of current studies and suggested areas for future research to address these gaps, providing a more nuanced understanding of the field.
Methodological Details:
Issue Example: We agree with the need to provide more details on the methodologies of mapping techniques used in neurosurgery. We expanded the section on direct electrical stimulation (DES) to include technical specifics such as stimulation parameters, site selection criteria, and protocols for patient safety.
Minor Comments:
Consistency in Terminology:
Issue Example: We checked the manuscript consistency in the use of technical terms and abbreviations, defining them at their first occurrence and maintaining consistent terminology for readability and comprehension.
Comments on the Quality of English Language:
We also reviewed and revised the manuscript to ensure clarity and smoother transitions between sections, as well as to simplify technical jargon where necessary.
We appreciate your thorough review and valuable suggestions, which will undoubtedly improve the quality of our manuscript. If you have any further comments or concerns, please do not hesitate to let us know.